# Selection of Bacterial Strains for Control of Root-Knot Disease Caused by *Meloidogyne incognita*

**DOI:** 10.3390/microorganisms9081698

**Published:** 2021-08-10

**Authors:** Varvara D. Migunova, Natalia S. Tomashevich, Alena N. Konrat, Svetlana V. Lychagina, Valentina M. Dubyaga, Trifone D’Addabbo, Nicola Sasanelli, Anzhela M. Asaturova

**Affiliations:** 1A.N. Severtsov Institute of Ecology and Evolution, Russian Academy of Sciences, Leninsky Prospect 33, 119071 Moscow, Russia; 2Federal State Budgetary Scientific Institution, Federal Scientific Center of Biological Plant Protection (FSBSI FSCBPP), 350039 Krasnodar, Russia; nataliatomashevich@yandex.ru (N.S.T.); dubyaga608@mail.ru (V.M.D.); asaturovaanzhela@yandex.ru (A.M.A.); 3Federal State Budget Scientific Institution, Federal Scientific Centre VIEV (FSC VIEV) of RAS, Bolshaya Cheryomushkinskaya 28, 117218 Moscow, Russia; alenakonrat@vniigis.ru (A.N.K.); s.lychagina@vniigis.ru (S.V.L.); 4Institute for Sustainable Plant Protection, CNR, Via G. Amendola 122/D, 70126 Bari, Italy; trifone.daddabbo@ipsp.cnr.it (T.D.); nicola.sasanelli@ipsp.cnr.it (N.S.)

**Keywords:** *Bacillus velezensis*, *Meloidogyne incognita*, *Fusarium*, *Rhizoctonia*, root-knot nematodes, nematicidal activity, antagonistic activity, plant growth

## Abstract

Root-knot disease caused by *Meloidogyne incognita* leads to significant crop yield losses that may be aggravated by the association with pathogenic fungi and bacteria. Biological agents can be effectively used against the complex disease of root-knot nematode and pathogenic fungi. In this study, 35 bacterial strains were analyzed for their in vitro nematicidal, antagonistic and growth stimulation activities. Based on results from the in vitro assays, grow-box experiments on tomato and cucumber were carried out with the strain BZR 86 of *Bacillus velezensis* applied at different concentrations. Effects of *B. velezensis* BZR 86 on the development of root-knot disease were evaluated by recording root gall index, number of galls and number of eggs in egg masses. Application of *B. velezensis* BZR 86 noticeably decreased the development of root-knot disease on tomato and cucumber plants, as well as significantly increased growth and biomass of cucumber plants in accordance with bacterial concentration. This study seems to demonstrate that strain *B. velezensis* BZR 86 could be an additional tool for an environmentally safe control of root-knot disease on horticultural crops.

## 1. Introduction

Root-knot nematodes (RKNs) are agents of severe root-knot disease (RKND) on many crops [1]. In Russia alone crop losses from RKND caused by the root-knot nematode *Meloidogyne incognita* Kofoid and White (Chitw.) ranged between 20% and 80% in cucumber and tomato plants cultivated under greenhouse conditions [2]. Symptoms of disease can be aggravated by the presence of bacterial and fungal pathogens using paths of RKNs penetration into plant roots [3,4,5], which means that control of this complex disease requires a suppression of RKNs and plant pathogenic microorganisms in soil, as well as stimulation of plant growth.

One of the environmentally safe methods for plant protection from RKND is application of microorganisms and/or products of their metabolism. Pesticides and growth regulators of microbial origin have proved their significant potential in sustainable agriculture and consequently in the development of green environment [6]. Microbial communities suppress RKNs, and genus *Bacillus* as a member of these communities functions as a biological agent, significantly decreasing the number of galls and egg masses of RKNs [7].

In particular, a significant role can be played by plant growth-promoting bacteria (PGPB), i.e., soil bacteria associated with plant roots reported for increasing plant resistance to biotic and environmental stresses and stimulating plant growth [8].

Application of PGPB has been repeatedly documented for reducing root-knot nematode infestations [9,10,11,12,13] and enhancing plant growth and yield at the same time [13,14,15,16]. RKND suppression by PGPB may involve different mechanisms such as competition, plant surface colonization, production of nematicidal and antimicrobial compounds (antibiotics, siderophores, hydrolytic enzymes, etc.), enhancement of host defense mechanisms [17,18,19,20].

Within PGPB, the most promising source of root-knot nematode control agents is represented by species of *Bacillaceae* family, widely occurring in soil and plant aerial parts and roots [21,22]. Most of available commercial products are formulations of *Bacillus subtilis* and *B. amyloliquefaciens* [23]. Moreover, *Bacillus firmus* I-1582 (Bf I-1582) and *B. amyloliquefaciens* FZB42 (now reclassified as a strain of *B. velezensis* [24]) have been approved for use against RKNs on vegetable crops in Europe [25]. Mechanisms of suppressiveness of these species to RKNs are related to egg colonization and degradation, as well as to the induction of a plant systemic resistance [26]. Moreover, a commercial formulation based on *B. velezensis* (Botrybel) is available as biofungicide [27].

The general aim of this study was to evaluate growth stimulation, nematicidal and antagonistic efficacy of bacterial isolates associated with rhizoshere and rhizoplane of plants and select the most promising strain for control of root knot disease. Its objectives were (i) to determine the effects of bacterial isolates on nematodes and phytopathogenic fungi: *Fusarium oxysporum*, *F. graminearum* and *Rhizoctonia solani*, (ii) to analyze the influence of bacterial isolates on development of plants, (iii) to find out if the selected strain can control the root knot disease.

## 2. Materials and Methods

### 2.1. Bacterial Strains 

All the bacterial strains used in this study were isolated from soil, rhizoplane and rhizosphere of plants from Krasnodar region (Russian Federation) (Appendix A). The isolation was done using the Warcup method and dilution technique [28]. The 35 strains tested in the in vitro assays were selected from the Bioresource Collection “State Collection of Entomoacariphages and Microorganisms” of Federal Scientific Center of Biological Plant Protection (Bioresource Collection of FSCBPP) according to their lipase, chitinase and protease activity (Appendix A) [29,30] as enzymatic activities may be involved in the control efficacy on plant parasitic nematodes [31]. In this research, we used the scientific equipment «Technological line for obtaining microbiological plant protection products of a new generation» (https://ckp-rf.ru/usu/671367/ (accessed on 1 July 2021)). Some of the isolates were identified by standard microbiological characterization [32] followed by 16S rRNA gene sequence analysis [33], full genome sequence was obtained for the strains that had demonstrated high nematicidal and antagonistic activities, namely: BZR 86 (https://www.ncbi.nlm.nih.gov/bioproject/PRJNA677970, (accessed on 12 November 2020)), BZR 277 (https://www.ncbi.nlm.nih.gov/bioproject/PRJNA677969, (accessed on 12 November 2020)) [34] and BZR 517 (https://www.ncbi.nlm.nih.gov/assembly/GCA_009683155.1#/def, (accessed on 12 November 2020)) [35]. The multiple alignment of concatenated amino acid sequences of 120 bacterial single-copy marker genes was carried out using the Genome Taxonomy Data Base (GTDB-Tk v.1.3.0 toolkit software) by RefSeq and Genbank genomes (U.S.A.) [36]. This multiple alignment was used to construct the maximum likelihood phylogenetic tree using PhyML v.3.3 [37], using default parameters. The level of support for internal branches was assessed using the Bayesian test in PhyML.

### 2.2. Antagonistic Activity against Phytopathogenic Fungi

Antagonistic activity of bacterial strains was determined by dual-culture plate method on potato glucose agar and King B medium [38]. A mycelial plug of *Fusarium oxysporum*, *F. graminearum* or *Rhizoctonia solani* was put in a Petri dish and a bacterial strain was plated at the distance of 6 cm from fungus. Control plates contained the fungus and bacterium alone. Cultures were incubated for 20 days at 28 °C. The growth of colonies was checked every day. The presence of sterile zone and its size, as well as fungal color, density and direction of mycelial growth were registered. Antagonistic activity was calculated according to the formula: % inhibition = [1 − (Fungal growth/Control growth)] × 100 [39].

### 2.3. In Vitro Nematicidal Activity 

The 35 bacterial strains were grown in liquid medium 925 [40], having the following composition: 3 g L^−1^ K_2_HPO_4,_ 1 g L^−1^, NaH_2_PO_4_, 1 g L^−1^ NH_4_Cl, 0.3 g L^−1^ MgSO_4_, 10 g L^−1^ sucrose, 2 g L^−1^ peptone, 1 L water. Fungivorous nematode *Paraphelenchus tritici* was used as model organism for the preliminary screening of nematicidal activity. Nematode population was grown on the fungus *Alternaria tenuis* in Petri dishes and then extracted by the Baermann funnel technique [41,42]. A 0.5 mL of the *P. tritici* water suspension, containing 50 nematode specimens, was added to each well of 24-well plates and then added with a 0.5 mL of each bacteria suspension. The test was done in five replicates. Nematode mortality was detected microscopically after 24 h. 

Bacterial strains’ activity on the second-stage juveniles (*J2*) of *M. incognita* was determined using bacterial suspensions grown in liquid medium 925 and their supernatants. Supernatants were obtained by centrifugation of bacterial suspension at 10,000 rpm for 10 min and did not contain any living bacterial cells. Bacterial suspension at concentration 10^8^ CFU mL^−1^ was diluted 10, 50, 100, 1000 and 10,000 times. A 0.5 mL amount of the obtained preparation of tested bacteria or supernatants was pipetted into 24-well plate with 0.5 mL of viable *J2* of *M. incognita* (100 ind.). Nematicidal effect was monitored after 24 and 48 h, after which nematodes were placed in sterile water for further 24 h to check for possible nematostatic effect. *J2* mortality was calculated according to the the Schneider Orelli’s formula: Corrected % Mortality = ([mortality % in treatment − mortality % in control]/[100 − mortality % in control]) × 100 [43]. The test was done twice in eight replicates. 

### 2.4. In Vitro Activity on Plant Growth

Thirty wheat seeds (cv. Raduga) were placed in Petri dishes with gauze, treated with 15 mL of bacterial suspensions and grown for 72 h at 25 °C. Effects of bacterial strains on plant growth was evaluated by the germination index (GI), i.e., the ratio between length of treated wheat seedlings and non-treated seedlings.

### 2.5. Pot Experiments

#### 2.5.1. Effects on Wheat Plants

Seeds of winter wheat (cv. Batko) not sterilized were soaked for 2 h in two-day bacterial cultures 10^9^ CFU mL^−1^ obtained by washing bacteria from Petri dishes followed by adding tap water to reach the volume of 50 mL. After 2 h, seeds were removed from bacterial suspensions and dried on filter paper. After 20–24 h, 30 seeds were sown in each of the three 0.45 L pots filled with sterilized sand. In total 90 seeds were planted, of which 80 germinated. The pots were stored in greenhouse at 24–28 °C and 11,000 Lux. The length of roots, height of stems and plant biomass were measured after 14 days. The testing of root length and stem height was done in 80 replicates. As for the biomass of the plants, we measured it by weighing each of the three pots because of the small size of individual plants. The experiment was performed twice.

#### 2.5.2. Effects on RKND and Plant Growth in Grow-Box Experiment

Strain BZR 86 was selected for the experiments in soil infested by RKN. A 1:1 mixture of peat and sand was poured in 0.18 L plastic pots, which were then sown with three–week-old cucumber seedlings cv. Kurazh (parthenocarpic) with three true leaves; the second test was done with 68-day-old tomato seedlings cv. Balkonnoe chudo. Two separated pot experiments with cucumber and tomato plants were conducted under grow-box conditions in All-Russian Scientific Research Institute for Fundamental and Applied Parasitology of Animals and Plants (Moscow, Russia). A *M. incognita* population, originally collected in Krasnodar region (Russian Federation) was reared on tomato (cv. Balkonnoe chudo) roots for 70 days, after which nematode egg-masses were picked and incubated in sterile water at 25 °C. The emerged *J2* were pipetted into each pot at a density of 150 *J2*/pot. A bacterial culture of *B. velezensis* BZR 86, resulted as the most effective strain in the in vitro screenings, was prepared by fermentation in flasks in the medium 925. The culture was maintained for 45 h at 29 °C under shaking (190 rpm), as to reach about 10^8^–10^9^ CFU mL^−1^ concentration. A 50 mL volume of bacterial suspension diluted in tap water was added to each pot, as to reach three different test concentrations: maximal: (3–7) × 10^6^ CFU mL^−1^ of soil substrate, medium: (3–7) × 10^5^ CFU mL^−1^ of soil substrate and minimal: (3–7) × 10^4^ CFU mL^−1^ of soil substrate. Concentrations of bacterial suspensions were determined by counting the colony forming units (CFU) on Luria–Bertrani agar (Sigma). Non-treated soil, either infested and non-infested with *M. incognita*, and infested soil treated with the chemical standard Phytoverm (avermectin C, 2 gL^−1^) were used as controls. Seven replicates were provided for each treatment. 

Pots were maintained at 25 °C under grow-box conditions for two months. At the end of the experiments, the height, number of leaves, ovaries, weight of aerial part and root biomass and root volume (as water displacement) were measured on each plant. Effects of *B. velezensis* BZR 86 on RKN infestation were determined by estimating the root gall index (RGI) according to a 0–5 scale, in which: 0 = no galls; 1 = 0.1%–10%; 2 = 11%–35%; 3 = 36%–70%; 4 = more than 70%; 5 = dead plant [41], as well as by microscopically counting number of eggs in egg masses.

### 2.6. Statistical Analysis

Data from the experiments were subjected to analysis of variance (ANOVA) and means compared by Duncan’s multiple range test (*p* < 0.05). All statistical analyses were performed using Microsoft Excel (standard deviation) and Statistica Version 13.5.0.17.T. (ANOVA, normality of data and Duncan’s multiple range test).

## 3. Results

### 3.1. Characterization of Bacterial Strains

Nematicidal activity varied from 0% to 100% among the 35 bacterial strains, though 46% showed a nematicidal activity higher than 85% (Table 1). The antagonistic activity against phytopathogenic fungi varied from 0 to 58% though only 14 of the studied strains showed a simultaneous antagonistic activity against *Fusarium graminearum*, *F. oxysporum* and *Rhizoctonia solani* higher than 30%. GI values ranged from 0.22 to 1.23, but only 28% of strains demonstrated a growth stimulating effect (GI > 1).

Only four strains simultaneously presented a high nematicidal effect (100%), an antagonistic activity against *F. graminearum, F. oxysporum* and *R. solani* (>30%) and a growth stimulation effect (GI > 1). These four strains were *Bacillus* species identified as BZR 86, BZR 623, BZR 261 and BZR 441. 

In the pot experiment on wheat, significant effects on plant root length and height occurred for strains BZR 441, BZR 623 and BZR 261 (Appendix A, Table 2). These effects were both negative (strain BZR 441) and positive (strains BZR 623 and BZR 261). Conversely, no significant difference from the control was found for BZR 86. 

Phylogenetic tree based on the whole genome sequencing of 120 conserved marker genes shows that strain BZR 86, selected according to results from the in vitro screenings, distinctly clusters with *B. velezensis* strain NRRL B-41580 (*B. velezensis* GCF 001461 825.1 on the phylogenetic tree); the average nucleotide identity (ANI) value is 97.59 % (Figure 1). The strain BZR 86 is related more closely to *B. siamensis* and *B. amyloliquefaciens* than to *B. subtilis* (as was determined by 16S rRNA).

Figure 2 presents nematicidal activity of *B. velezensis* BZR 86 against *J2* of *M. incognita*. LD_50_ and LD_99_ of *J2* were 0.8 × 10^6^ and 6.8 × 10^6^ CFUmL^−1^, respectively. 

Its antagonistic activity against *F. graminearum, F. oxysporum* and *R. solani* was 45.9%, 32.4% and 45.8%, respectively (Table 1, Figure 3).

### 3.2. Influence of B. velezensis BZR 86 on development of the Root Knot Disease

In the non-treated control, an attack of *M. incognita* led to a moderate damage both on cucumber and tomato plants, as root gall index varied from 2.6 to 2.8 (Table 3 and Table 4). The experiments showed that *B. velezensis* BZR 86 influenced the development of root-knot disease on cucumber and tomato plants, reducing root gall index, number of galls and eggs in egg masses when applied to soil substrate without significant differences from chemical control Phytoverm (Table 3 and Table 4). 

In the experiment on cucumber, the number of galls and the root gall index were significantly lower than at 7 × 10^4^ and 7 × 10^6^ CFU mL^−1^ of substrate concentrations of *B. velezensis* BZR 86. All three concentrations of *B. velezensis* BZR 86 significantly reduced the number of eggs in egg masses (Table 3). The application of bacteria to soil substrate influenced the growth and development of cucumber plants. At the end of the experiment, considerable difference (almost double) in height was detected between the application of *B. velezensis* BZR 86 at 7 × 10^6^ CFU mL^−1^ of substrate concentration and control, Phytoverm and non-infested soil. Moreover, at the highest concentration of *B. velezensis* BZR 86, the number of plant leaves was significantly greater compared to Phytoverm and non-infested control. 

By application of bacterium at the highest concentration, the biomass of the aerial part of cucumber plants increased twice compared with control, chemical standard and that at the lowest concentration (Table 5).

In the experiment on tomato, application of *B. velezensis* BZR 86 almost completely eliminated the symptoms of RKND at all concentrations. Only single galls were formed on roots. At the lowest concentration of *B. velezensis* BZR 86 (3 × 10^4^ CFU mL^−1^ of substrate), the gall numbers tended to increase, but the difference from other bacterial concentrations was not statistically significant (Table 4).

Application *B. velezensis* BZR 86 at 3 × 10^6^ and 3 × 10^5^ CFU mL^−1^ of substrate concentrations resulted in a significant increase in tomato root system biomass compared to chemical standard Phytoverm and non-infested soil, whereas this effect was absent at the lowest bacterium concentration. At the medium concentration of bacterium (3 × 10^5^ CFU mL^−1^ soil), tomato root volume was significantly greater than at the lowest concentration and also than Phytoverm. It exceeded that parameter for the variant without RKN by four times (Table 4).

## 4. Discussion

The problem of root-knot disease on cucumber and tomato plants is really pressing in Russia, as most Russian farmers use soil substrates frequently infested by root-knot nematodes rather than hydroponics. At present, there are no microbial products registered against root-knot disease in the Russian Federation [44]. Chemical pesticide Phytoverm does not help to solve the problem when plant roots are strongly affected by the disease resulting in premature death of plants. 

*Bacillus* species have been screened for nematicidal activity against RKNs [15,45,46], as well as for antifungal properties [47,48,49] and growth stimulation activity [50,51]. A number of studies documented a simultaneous activity of *Bacillus* strains against RKNs and fungi [19], as well as against RKNs or fungal phytopathogens, and its effect on plant growth enhancement [11,16,52]. In this study, we analyzed all these three parameters in the in vitro experiment, so as to select the most promising strain, i.e., *B. velezensis* BZR 86 isolated from winter wheat rhizosphere. 

Previous research has shown a significant influence of genus *Bacillus* on mortality of juveniles of the root-knot nematode *M. incognita* [15,43,44,45,46,53,54]. Among others, *B. velezensis* had nematicidal activity against *M. incognita* [11,55,56,57]. In vitro *B. velezensis* BZR 86 caused 98% mortality of *M. incognita J2* following a 24-h exposure to a 9 × 10^7^ CFU mL^−1^ bacterial density [55]. In our study the nematicidal effect of *B. velezensis* BZR 86 was also confirmed on cucumber and tomato in soil, in agreement with previous studies that reported a significant suppressiveness of *Bacillus* strains on root-knot nematode eggs, *J2* and root galls on cucumber [10,55], tomato [16,17,53,58], eggplant [15,59] and hendi [60]. The mechanisms underlying the RKN suppression by *Bacillus* strains can be different. Thus Burkett-Cadena et al. [17] suggested the production of antibiotic metabolites as responsible for soilborne pathogen suppression by *B.subtilis* GB03, while an induced systemic resistance (ISR) has been suggested as the main mechanism of biocontrol activity of *B. amyloliguefaciens* strain FZB42 [61]. 

Induced systemic resistance could also be a possible mechanism of *B. velezensis* BZR 86 suppressiveness on *M. incognita*, as some *Bacillus* strains were found to release jasmonic acid, known for reducing *M. incognita* infestation of tomato and cowpea by triggering plant defense against the root-knot nematode [61,62,63,64]. In good agreement, Toral et al. [63] reported significant increases in salicylic and jasmonic acid levels in strawberry plants treated with *Bacillus velezensis* XT1. 

*Bacillus* strains were also found to produce nematicidal and antimicrobial compounds, including antibiotics, cyclic lipopeptides, polyketides and bacteriocins [23,65]. Since we observed a significant nematicidal effect of *B. velezensis* BZR 86 against *M.incognita* also in the in vitro assays, we may suppose that secondary metabolites of this strain could have a role in controlling RKND also in soil. Studies on genome of *B. velezensis* FZB 42 revealed that 13 gene clusters are responsible for the synthesis of predicted antimicrobial metabolites or volatile compounds [66].

Plant growth stimulation by *B. velezensis* BZR 86 observed in our experiments on cucumber and tomato is in good agreement with growth increase observed on olive trees treated with *B. velezensis* OEE1 [67]. Moreover, new isolated *Bacillus* strains were already stated for enhancing crop growth and productivity [65] and improving soil health [50,68,69]. Some bacilli are also known to fix N_2_ and soluble phosphate [70,71], thus promoting circulation of plant nutrients and, consequently, increasing crop growth and yield. Genes that contribute to plant growth promotion provide the possibility to use *B. velezensis* as a biofertilizer. It is known that bacilli produce plant growth promoting phytohormons (cytokinin, auxin) and volatile organic compounds (aceton (3-hydroxy-2-butanone), 2,3,-butanediol) [52,72,73]. The significant increase in the root volume and root biomass of cucumber and tomato plants may be also due to locally reduced ethylene concentrations and increased assimilation of metal ions, such as iron, through the activation of plant’s own iron acquisition mechanisms by members of the *Bacillaceae* [21]. Moreover, Qin et al. [68] documented a significant effect of *B. amyloliguefaciens* LS-60 on the structure of bacterial community associated with cucumber seedling. This change of community structure resulted in the dominance of genera *Bacillus*, *Rhodanobacter, Paenibacillus*, *Pseudomonas*, *Nonomuraea* and *Agrobacterium*, known for great impacts on soil nutritional composition, mineral metabolism and antibiotic production, thus providing an increased content of available nitrogen, phosphorus and potassium in soil substrate. 

The number of bacteria in soil varies from 10^7^ to 10^9^ CFU per gram of soil [21]. We inoculated the bacterial formulations into soil substrate at concentrations far lower (10^4^–10^6^ CFU mL^−1^ of soil substrate) in order to minimize the effect on the ecosystem. However, future investigations are needed to analyze the mechanisms involved in the interaction among rhizospheric bacteria, plants and other components of soil ecosystem. There was no significant direct correlation between the concentration of bacterial formulations and development of RKND. 

Our own results and data of other researchers [29,47,48,49,64,67,74,75,76,77] demonstrated a strong antagonistic activity of *B. velezensis* to the pathogenic fungi *Fusarium graminearum*, *F. oxysporum* and *R. solani,* thus confirming this species as a very promising biocontrol agents [24,78,79]. Practical value of this antagonism has been proved by the use of *B. velezensis* in BioYield commercial formulation applied in management of soil-born pathogens and *M. incognita* on tomato [11].

The data obtained in this and further research will contribute to selection and comprehensive study of new biocontrol strains. They may form the basis for plant protection against plant parasitic nematodes as an alternative control measure. This also implies serious commercial potential, as the demand for such formulations is growing while global supply remains insufficient.

## 5. Conclusions

Selection of bacterial strains for control of RKND should be comprehensive and include analysis of nematicidal, fungicidal and growth stimulation activities. A number of bacterial strains with these characteristics are already present in the Bioresource Collection of FSCBPP that should be considered a valuable source of new innovative products for sustainable agricultural systems.

Bacterial strain *B. velezensis* BZR 86 showed to have a multiple effect on the complex of plants, RKNs and phytopathogenic microorganisms and, therefore, could be a potential candidate for the production of new biostimulants with a side suppressive activity on RKND. However, mechanisms of this multiple activity should be further investigated in detail, as to improve effects by optimizing techniques and timing of application. 

## Figures and Tables

**Figure 1 microorganisms-09-01698-f001:**
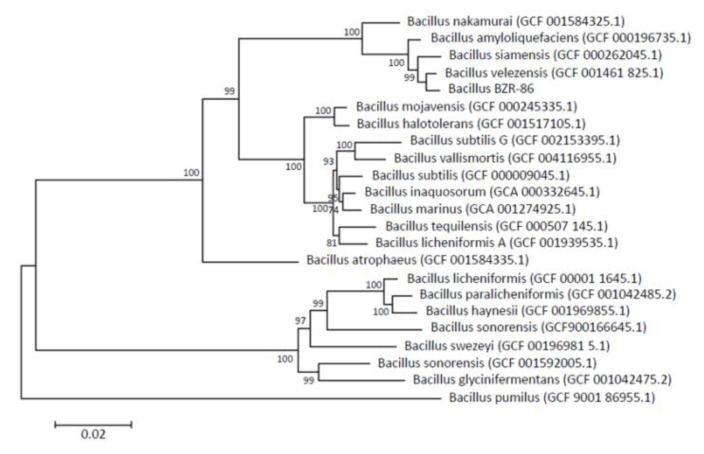
Maximum likelihood phylogenetic tree constructed using amino acid sequences of 120 conserved marker genes. The tree was constructed using PhyML v.3.3.

**Figure 2 microorganisms-09-01698-f002:**
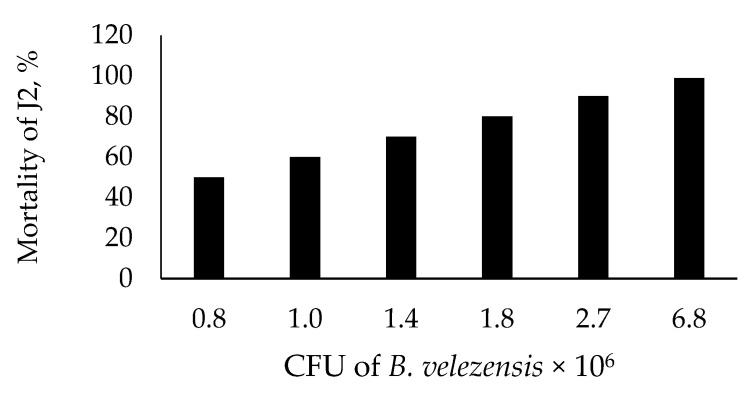
Nematicidal effect of *Bacillus velezensis* BZR 86 on *J2* of *Meloidogyna incognita.*

**Figure 3 microorganisms-09-01698-f003:**
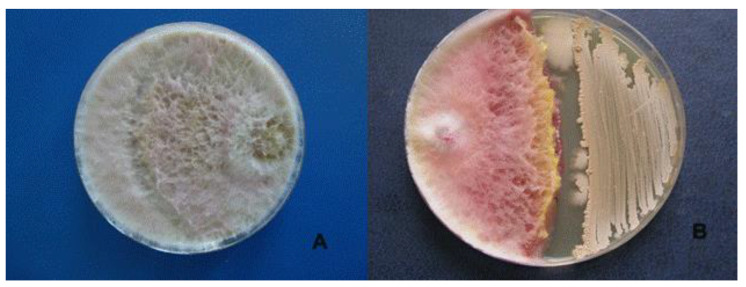
Antagonistic effect of *Bacillus velezensis* BZR 86 on *Fusarium graminearum. (***A**)—control, (**B**)—*Bacillus velezensis* BZR 86.

**Table 1 microorganisms-09-01698-t001:** Germination index, nematicidal and antagonistic effects of bacterial strains selected from the Bioresource Collection of FSCBPP.

Strain ^1^	Nematicidal Activity (%)	GI ^2^	Antagonistic Activity on the 15th Day (%)		
			*F. oxysporum*	*R. solani*	*F. graminearum*
BZR 18	92 ± 3.1 ^3^ ij ^4^	0.84 ± 0.06 efg	0 ± 0 k	0 ± 0 k	53.9 ± 2.2 bcd
BZR 59	63 ± 0 hi	0.86 ± 0.23 efgh	0 ± 0 k	28 ± 1.3 ij	52.9 ± 1.4 cd
BZR 86	100 ± 5 j	1.18 ± 0.06 mn	32.4 ± 2 f	45.8 ± 0.8 bcd	45.9 ± 0.8 h
BZR 148	4 ± 0 ef	0.78 ± 0.08 def	42.2 ± 0.8 a	48.4 ± 0.8 a	41.5 ± 1.4 ij
BZR 187	100 ± 0 j	0.94 ± 0.08 ghij	40 ± 1.3 c	43.1 ± 1.5 e	32.2 ± 0.8 mn
BZR 241	96 ± 0 j	0.87 ± 0.08 efghi	41.8 ± 0.8 abd	43.1 ± 3.4 e	41 ± 0.8 ij
BZR245 F	12 ± 0.6 bcd	0.44 ± 0.09 b	25.8 ± 2.0 i	25.8 ± 2.0 j	28 ± 0.8 o
BZR 261	100 ± 0.5 j	1.10 ± 0.09 klmn	36.4 ± 0.8 e	46.7 ab	39.5 ± 0.8 j
BZR 277	92 ± 0 ij	0.22 ± 0 a	40.9 ± 0.8 abcd	44 ± 2.7 cde	42.9 ± 1.4 i
BZR 337	70 ± 0 g	0.67 ± 0.05 cd	0 ± 0 k	0 ± 0 k	52.5 ± 0.8 cd
BZR 348	17 ± 0 cd	0.73 ± 0.10 cde	24 ± 2.3 j	46.7 ab	35.9 ± 2.5 kl
BZR 367	0 ± 0 a	1.00 ± 0 hijkl	25.3 ± 2.3 ij	28.4 ± 2.0 i	30.8 ± 2.2 n
BZR 413	82 ± 0 ij	1.20 ± 0.07 n	0 ± 0 k	0 ± 0 k	31.8 ± 0.8 n
BZR 416	16 ± 1.7 cd	0.93 ± 0.10 ghij	25.8 ± 0.8 i	25.8 ± 0.8 j	28.3 ± 0.8 o
BZR 417	12 ± 0 de	1.02 ± 0.05 ijkl	39.6 ± 0.8 c	40 ± 3.5 f	39.9 ± 1.4 j
BZR 430	10 ± 0 bc	1.02 ± 0.09 ijkl	0 ± 0 k	0 ± 0 k	42.4 ± 0.8 i
BZR 436	7 ± 0 cd	0.84 ± 0.29 efg	28 ± 1.3 gh	32 ± 2.3 h	50 ± 1.4 ef
BZR 441	100 ± 0 j	1.06 ± 0.08 jklm	39.1 ± 2.0 c	43.6 ± 0.8 de	54 ± 1.4 bc
BZR 455	100 ± 0 j	1.13 ± 0.07 lmn	0 ± 0 k	0 ± 0 k	58.1 ± 2.2 a
BZR 462	93 ± 11.5 ij	1.00 ± 0 hijkl	42.2 ± 0.8ab	47.1 ± 0.8 ab	36.9 ± 0.8 k
BZR 472	92 ± 3.5 ij	0.68 ± 0.07 cd	0 ± 0 k	0 ± 0 k	48.5 ± 1.4 fg
BZR 480	100 ± 0 j	1.00 ± 0 hijkl	0 ± 0 k	0 ± 0 k	34.3 ± 1.4 lm
BZR 512	100 ± 0 j	0.63 ± 0 c	0 ± 0 k	0 ± 0 k	48 ± 0.8 fg
BZR 517	80 ± 0 hi	0.67 ± 0.17 cd	36.9 ± 0.8 e	43.6 ± 0.8de	52.5 ± 0.8 cd
BZR 519	100 ± 0.5 j	0.94 ± 0.08 ghij	40 ± 0 cd	46.2 ± 4.1 abc	54.9 ± 1.6 bc
BZR 523−1	84 ± 4.2 hi	0.89 ± 0 fghi	26.7 ± 0 hi	43.6 ± 0.8 de	51.5 de
BZR 523-2	16 ± 0 cd	0.89 ± 0.11 fghi	0 ± 0 k	0 ± 0 k	48.5 ± 1.4 fg
BZR 528	100 ± 0 j	0.89 ± 0.06 fghi	0 ± 0 k	0 ± 0 k	52.5 ± 1.4 cd
BZR 538	11 ± 2.5 bc	0.75 ± 0 cdef	0 ± 0 k	0 ± 0 k	45.3 ± 1.4 h
BZR 623	100 ± 0.6 j	1.10 ± 0.09 klmn	40.4 ± 0.8 bcd	48.4 ± 0.8a	41 ± 0.8 ij
BZR 658	100 ± 0 j	1.23 ± 0.09 n	0 ± 0 k	0 ± 0 k	39.5 ± 0.8 j
BZR 673	9 ± 0 de	0.96 ± 0.06 ghijk	42.2 ± 2.0 ab	47.1 ± 0.8 ab	47.1 ± 1.4 gh
BZR 854	36 ± 0 f	0.86 ± 0.21 efgh	39.1 ± 0.8 c	48 ± 1.3 ab	55.4 ± 0.8 b
BZR 862	96 ± 0 j	0.78 ± 0 def	28.4 ± 0.8 g	34.7 ± 1.3 g	41.9 ± 0.8 ij
BZR 873	77 ± 19.6 gh	1.23 ± 0.09 n	0 ± 0 k	0 ± 0 k	52.9 ± 1.4 cd

^1^ Accession number of bacteria in the Bioresource Collection of FSCBPP; ^2^ GI—germination index; ^3^ average ± SD of eight replicates; ^4^ data flanked in each column by the same letters are not statistically different according to Duncan’s multiple range test (*p* = 0.05).

**Table 2 microorganisms-09-01698-t002:** Effect of bacterial strains on growth and biomass of winter wheat plants (cv. Batko) in the pot experiment.

Treatment	Plant Height (cm)	Root Length (cm)	Weight of Dry Biomass (g)	
			Aerial Parts	Roots
Control	14.6 ± 1.8 ^1^ a ^2^	18.3 ± 3.7 a	0.11 ± 0 a	0.15 ± 0.01 d
BZR 441	14.8 ± 1.7 a	16.4 ± 2.9 b	0.11 ± 0.01 a	0.13 ± 0.01 bcd
Control	14.9 ± 1.6 a	14.6 ± 2.1 c	0.11 ± 0.01 a	0.12 ± 0.02 ab
BZR 623	15.8 ± 1.9 b	14.5 ± 3.1 c	0.12 ± 0.01 a	0.14 ± 0.01 cd
Control	14.8 ± 1.9 a	14.6 ± 3.2 c	0.11 ± 0.01 a	0.10 ± 0.01 a
BZR 86	15.1 ± 1.7 a	14.5 ± 3.0 c	0.11 ± 0.01 a	0.13 ± 0.01 bcd
Control	14.8 ± 1.9 a	14.6 ± 3.2 c	0.11 ± 0 a	0.10 ± 0.01 a
BZR 261	14.7 ± 1.9 a	17.0 ± 3.6 b	0.11 ± 0 a	0.13 ± 0.01 bc

^1^ Each value is an average ± SD of six replicates (30 plants/replicate) from two independent experiments; ^2^ data followed by the same letters are not statistically different according to Duncan’s multiple range test (*p* = 0.05).

**Table 3 microorganisms-09-01698-t003:** Influence of *Bacillus velezensis* BZR 86 on the development of root-knot disease of cucumber (cv. Kurazh).

Treat	Galls/Root	Eggs/Egg Mass	Root Gall Index (1–5)
Control	218 ± 58.6 ^1^ c ^2^	316 ± 41.9 a	2.6 ± 1.1 b
Phytoverm	73 ± 13.2 ab	191 ± 19.0 b	1 ± 0.6 a
BZR 86, 7 × 10^6^ CFU mL^−1^	86 ± 16.8 ab	207 ± 36.1 b	1.4 ± 0.9 a
BZR 86, 7 × 10^5^ CFU mL^−1^	141 ± 31.8 bc	177 ± 24.4 b	1.8 ± 0.5 ab
BZR 86, 7 × 10^4^ CFU mL^−1^	65 ± 16.0 ab	224 ± 33.9 b	1.64 ± 0.7 a

^1^ Each value is an average ± SD of seven replicates. ^2^ Data flanked in each column by the same letters are not statistically different according to Duncan’s multiple range test (*p* = 0.05).

**Table 4 microorganisms-09-01698-t004:** The influence of *Bacillus velezensis* BZR86 on the growth and infestation of *Meloidogyne incognita* on tomato (cv. Balkonnoe chudo).

Treatment	Height (cm)	N° Leaves	Plant Biomass (g)		Root Volume (mL)	Root Gall Index (0–5)	Galls/Root
			Aerial Part	Roots			
Non-infested control	16.2 ± 2.4 ^1^ a ^2^	12 ± 1.0 ab	5.0 ± 1.6 a	1.0 ± 0 d	1.2 ± 0.4 c	-	-
Infested control	25.3 ± 2.3 b	15 ± 1.9 b	8.8 ± 2.2 ab	3.5 ± 1.0 ab	4.0 ± 0.8 b	2.8 ± 1.0 b	81 ± 10.1 b
Phytoverm	20.0 ± 3.4 ab	12 ± 0.9 a	9.2 ± 1.5 b	2.2 ± 0.8 c	2.8 ± 1.0 a	0.1 ± 0 a	3 ± 1.5 a
BZR 86, 3 × 10^6^ CFU mL^−1^	24.0 ± 3.9 b	12 ± 2.0 ab	10.8 ± 1.0 b	3.8 ± 1.0 b	3.5 ± 0.6 ab	0.1 ± 0.1 a	2 ± 1.7 a
BZR 86, 3 × 10^5^ CFU mL^−1^	23.2 ± 1.2 b	15 ± 3.7 ab	9.7 ± 2.0 b	3.5 ± 0.5 ab	4.0 ± 0.6 b	0.1 ± 0.1 a	1.2 ± 0.8 a
BZR 86, 3 × 10^4^ CFU mL^−1^	21.3 ± 5.3 ab	15 ± 2.9 ab	8.1 ± 1.2 ab	2.4 ± 1.0 ac	2.9 ± 1.1 a	0.3 ± 0.4 a	8 ± 2.7 a

^1^ Each value is an average ± SD of seven replicates. ^2^ Data flanked in each column by the same letters are not statistically different according to Duncan’s multiple range test (*p* = 0.05, *n* = 7).

**Table 5 microorganisms-09-01698-t005:** The influence of *Bacillus velezensis* BZR86 on the growth and development of cucumber plants (cv. Kurazh).

Treatment	Height (cm)	N° Leaves	Biomass Weight (g)		Root Volume (mL)
			Aerial Parts	Roots	
Non-infested control	51 ± 10.9 ^1^ a ^2^	22 ± 4.1 a	10.9 ± 1.2 ab	4.2 ± 1.5 a	7.6 ± 2.2 b
Infested control	48 ± 10.4 a	22 ± 5.7 ab	6.3 ± 2.0 a	4.1 ± 2.1 a	3.1 ± 1.6 a
Phytoverm	44 ± 8.7 a	22 ± 4.2 a	6.1 ± 3.8 a	3.4 ± 1.7 a	6.0 ± 2.4 ab
BZR 86, 7 × 10^6^ CFU mL^−1^	92 ± 4.2 b	27 ± 1.8 b	12.9 ± 3.1 b	6.6 ± 2.1 a	6.0 ± 2.8 ab
BZR 86, 7 × 10^5^ CFU mL^−1^	57 ± 12.7 a	24 ± 3.2 ab	9.6 ± 2.6 ab	7.1 ± 1.5 a	6.4 ± 2.4 ab
BZR 86, 7 × 10^4^ CFU mL^−1^	50 ± 13.7 a	22 ± 3.4 ab	6.9 ± 2.8 a	4.7 ± 1.7 a	5.4 ± 1.5 ab

^1^ Each value is an average ± SD of seven replicates. ^2^ Data flanked in each column by the same letters are not statistically different according to Duncan’s multiple range test (*p* = 0.05, *n* = 7).

## Data Availability

Data is contained within the article or Appendix A.

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
