# Peer review of "Selection of Bacterial Strains for Control of Root-Knot Disease Caused by Meloidogyne incognita"

_microorganisms, 2021, doi:10.3390/microorganisms9081698_

Round 1

Reviewer 1 Report

Quite interesting paper concerning on bacterial strains selection to control root-knot Meloidogyne incognita, one of the most economically important plant-parasitic nematode. The obtained results suggest that selected bacterial strains exhibit antagonistic activity against pathogenic nematodes and fungi.

Some detailed comments below.

The authors should explain why the initial selection of the bacterial strains was based on their enzymatic activity. It is not obvious.

L79. Why have only certain strains of bacteria been identified? What was the selection criterion?

L96-97. There is no logic in this sentence. If the area is sterile, there should be no changes in color, size, etc.

L164. What statistical analyzes were made in Microsoft Excel? This software has quite limited possibilities to carry out statistical analyzes. How did authors check the normality of data distribution, equality of variance?

Tables and figures: no standard deviation/error indicated. If the result was 0 or 100% - these data should be excluded from statistical analysis (no variation). The highest or lowest scores should be marked with “a”.

Why the results of the antifungal activity of bacteria strains (Table 1) were not compared statistically? Table 2: which results were compared? In my opinion, there should be pairwise comparisons performed, not all groups.

Figure 1 is not necessary. GenBank accessions numbers mentioned in the text would be sufficient for bacteria identification.

Figure 2: what are the numbers above the bars? It should be removed.

The discussion should be shortened. Authors should focus on the obtained results and their comparison with the literature. No attempts were made to explain mechanisms of promoting or antagonistic activity of selected bacteria strains, so there is no need to discuss this in the article.

Author Response

Response to Reviewer 1 Comments

The authors should explain why the initial selection of the bacterial strains was based on their enzymatic activity. It is not obvious.

We have added explanation:

as enzymatic activities may be involved in the control efficacy on plant parasitic nematodes [30].

L79. Why have only certain strains of bacteria been identified? What was the selection criterion?

full genome sequence was obtained for the strains that had demonstrated high nematicidal and antagonistic activities

L96-97. There is no logic in this sentence. If the area is sterile, there should be no changes in color, size, etc.

We have changed the sentence.

L164. What statistical analyzes were made in Microsoft Excel? This software has quite limited possibilities to carry out statistical analyzes. How did authors check the normality of data distribution, equality of variance?

Data were recorded in Microsoft Excel just to have a simple and
immediate method to calculate Average and Standard Deviation (SD). Of course, the observation of the reviewer is right because
to analyze data from experiments are normally used statistical programs
(as we did). However, many statistical analyses can be carried out by
Microsoft Excel although its use can be complicated and no easy.
We used   “Statistica” (Version 13.5.0.17.T) which allows to
subject data to analysis of variance (ANOVA) when they are normally
distributed and to compare means by different tests (Duncan’s Multiple
Range Test or Last Significant Difference’s Test or other). Data were
verified for their normal distribution (Kolmogorov-Smirnov’s Test). On
the contrary, if the distribution of the variable of interest is not
normal we are forced to use non-parametric tests such as Wilcoxon,
Mann-Whitney or Kruskal-Wallis.

Tables and figures: no standard deviation/error indicated. If the result was 0 or 100% - these data should be excluded from statistical analysis (no variation). The highest or lowest scores should be marked with “a”.

We have added the standard deviation in the tables.

Why the results of the antifungal activity of bacteria strains (Table 1) were not compared statistically? Table 2: which results were compared? In my opinion, there should be pairwise comparisons performed, not all groups.

The results of the antifungal activity of bacterial strains were compared statistically.

Table 2: The observation of the reviewer 1 is right. However, the Duncan’s
Multiple Range Test allows to verify the statistical differences among
the different isolates and their controls. So, this statistical
comparison method allows to put in evidence not only the difference
between an isolate and its control but also the differences among all
bacterial isolates. So, the “t” Student test (pairwise comparison) is
not needed.

Figure 1 is not necessary. GenBank accessions numbers mentioned in the text would be sufficient for bacteria identification.

We consider Figure 1 important because it demonstrates that our strain BZR 86 belongs to B. velezensis and is related more closely to B.siamensis and B.amyloliquefaciens than to B.subtilis (as was determined by 16S rRNA).

Figure 2: what are the numbers above the bars? It should be removed.

We deleted them.

The discussion should be shortened. Authors should focus on the obtained results and their comparison with the literature. No attempts were made to explain mechanisms of promoting or antagonistic activity of selected bacteria strains, so there is no need to discuss this in the article.

It is not quite clear which mechanisms are meant by the Reviewer. We refer to such mechanisms in order to  explain our results.

We are grateful to the Reviewer for the valuable comments and proposal.

Reviewer 2 Report

The manuscript describes the use of bacterial isolates in biocontrol of Meloidogine incognita that causes root-knot disease. Thirty-five bacterial strains were isolated from the soil, rhizoplane, and rhizosphere of plants. They were tested for lipase, protease and chitinase activity and were identified by standard microbiological characterization followed by 16S rRNA gene sequence analysis or full genome sequence. Furthermore, antagonistic and nematicidal activities were applied. In addition, in vitro wheat growth and greenhouse pot experiment were performed to study the effect of isolates. Finally, statistical analyses were made using Excel and Statistica 13.5.0.17 with Duncan’s multiple range tests of differences.

The manuscript is well organized, and each of the sections is well and in detail developed according to the Instructions to the authors. The literature is well synthesized in the Introduction and could be accepted in the present form, but it will be good to use more recent references.

The questions set by the authors were successfully answered. The methodology is clearly explained and could be reproduced easily. The study possesses some novelty expressed in the isolation and identification of bacteria common for soils.

The results are clear and easy to understand. The discussion combines the results very well with the data from the literature in applying beneficial populations, expressing different abilities such as plant growth improvement, root-knot disease suppression, the existence of antagonistic effects, etc.

My recommendations are on the introduction and discussion, which need to be supplemented and clarified.

Author Response

We are very grateful to the reviewer for his/her comments and proposal.

We added and modified the text of the article.

Reviewer 3 Report

The work of Migunova et al. is focused on the screening of new bacterial strains active in the biocontrol of the plant parasitic nematode Meloidogyne incognita and of phytopathogenic fungi.

There are two points that I consider to be very critical in this paper:

1) the lack of novelty and 2) the validity of experiments based on a low number of experimental replicates (three replicates to determine the effect of bacterial strains on the growth of wheat in pots).

These main issues are associated with a number of other gaps that need to be filled.

  • It would be very nice, in a paper submitted to a journal focused on microbiology, showing the taxonomical identification of the bacterial strains, not only the strain label.
  • All the data reported in the table should be added with the standard deviation.
  • The use of the language must be improved.
  • A number of typos occur in the paper:

Line 34 the first reference is 1

Line 35-36 : ranged between 20 and 80% in cucumber and tomato plants cultivated under greenhouse conditions

Line 38: change needs to requires

Line 41-44: the sentence is too long. Please rephrase

Line 54-56: I think it would be useful to specify which advantages strains belonging to Bacillus genus can provide

Line 72-77: I suggest the Authors to add a brief explanation of the sampling method and of the procedure followed in order to discriminate bulk soil, rhizosphere and rhizosphere compartments.

Line 77: the physiological traits on which the Authors based the strain selection should be improved by characterizing other metabolic activities such as the synthesis of auxins, siderophores and the capability to solubilize phosphate. This is important since the Authors assessed the plant growth promotion activities of the strains.

Line 127-131: Were the seeds and the sand sterilized?

Line 127: Please, state clearly the density in CFU/ml of the bacterial inoculant.

Line 131: three replicates are not enough for a robust statistical analysis. Even more, since the pots were filled with sand I expect the Authors use a nutrient solution for growing plants. Please, add this information. What about the density of the bacterial strain on the seed after bacterial inoculation? What about the bacterial density on/in the root at the end of the experiment?

Line 182, 206, 214, 222, 226, 237, 267, 269, 270, 276, 279…. : A space is missing between the genus and the species 

Line 183, 259, 261, 272, 274, 285, 294 …..307, 308: Bacillus in italics

Line 221-231: I think the Authors should modify the sentence and improve the use of the language.

Line 222: change smaller to lower than

Line 231: change two times to twice

Line 240: a “,” is missing before “but”

Line 269: change concentration to bacterial density

Line 280: change excrete to release

Line 296: and solubilize phosphate

Line 312-314: I sincerely think the bacterial density used by the Authors is too low to outcompete with the resident microflora, especially in field conditions.

Author Response

Response to Reviewer 3

The work of Migunova et al. is focused on the screening of new bacterial strains active in the biocontrol of the plant parasitic nematode Meloidogyne incognita and of phytopathogenic fungi.

There are two points that I consider to be very critical in this paper:

  • the lack of novelty and 2) the validity of experiments based on a low number of experimental replicates (three replicates to determine the effect of bacterial strains on the growth of wheat in pots).
  • In our opinion, the novelty of this research consists in the approach to the selection of bacterial strains for control of root-knot disease. The use of this complex approach is not reflected in literature, hence our assumption that it is new. Secondly, the tested isolates are new for the science; they have considerable potential for creating new bacterial formulations to control plant parasitic nematodes in sustainable agriculture.

2.) We have introduced changes in the text of the Paper to clarify the point.

These main issues are associated with a number of other gaps that need to be filled.

  • It would be very nice, in a paper submitted to a journal focused on microbiology, showing the taxonomical identification of the bacterial strains, not only the strain label.

We agree with this statement, however, for the purposes of this research we needed to identify only 16S rRNA, which is not sufficient for full taxonomic identification.

  • All the data reported in the table should be added with the standard deviation.

We have added the standard deviation.

  • The use of the language must be improved.
  • A number of typos occur in the paper:

Line 34 the first reference is 1

Line 35-36 : ranged between 20 and 80% in cucumber and tomato plants cultivated under greenhouse conditions

Line 38: change needs to requires; we have changed it.

Line 41-44: the sentence is too long. Please rephrase

The sentence has been rephrased; now it includes information about the advantages the strains belonging to Bacillus genus can provide.

Line 54-56: I think it would be useful to specify which advantages strains belonging to Bacillus genus can provide

Line 72-77: I suggest the Authors to add a brief explanation of the sampling method and of the procedure followed in order to discriminate bulk soil, rhizosphere and rhizosphere compartments.

The isolation was done using the Warcup method and dilution technique [28].

Line 77: the physiological traits on which the Authors based the strain selection should be improved by characterizing other metabolic activities such as the synthesis of auxins, siderophores and the capability to solubilize phosphate. This is important since the Authors assessed the plant growth promotion activities of the strains.

We are grateful for the recommendation. We have been planning to analyze these activities at the next stage of our research project.

Line 127-131: Were the seeds and the sand sterilized?

The sand was sterilized, the seeds were not. We have added this to the text.

Line 127: Please, state clearly the density in CFU/ml of the bacterial inoculant.

Seeds of winter wheat (cv. Batko), not sterilized were soaked for 2 hours in two-day bacterial cultures 109 CFU ml-1

Line 131: three replicates are not enough for a robust statistical analysis. Even more, since the pots were filled with sand I expect the Authors use a nutrient solution for growing plants. Please, add this information. What about the density of the bacterial strain on the seed after bacterial inoculation? What about the bacterial density on/in the root at the end of the experiment?

We have explained above the point about the replicates number. There was no nutrient solution. We did not measure bacterial density either on the seed or on the root.

Line 182, 206, 214, 222, 226, 237, 267, 269, 270, 276, 279…. : A space is missing between the genus and the species 

Line 183, 259, 261, 272, 274, 285, 294 …..307, 308: Bacillus in italics

We have done it.

Line 221-231: I think the Authors should modify the sentence and improve the use of the language.

Line 222: change smaller to lower than

Line 231: change two times to twice

Line 240: a “,” is missing before “but”

Line 269: change concentration to bacterial density

Line 280: change excrete to release

Line 296: and solubilize phosphate

We have modified.

Line 312-314: I sincerely think the bacterial density used by the Authors is too low to outcompete with the resident microflora, especially in field conditions.

It may be so, but this requires further research.

Thank you very much indeed for your careful analysis and valuable recommendations.

Reviewer 4 Report

Fine article with logical concept and clearly formulated results, I can recommend its publication after some minor adjustment:

What was the unique scientific equipment used (lines 77 – 79), please write more about it.

Why were obtained genome sequences from three bacterial strains; what was the reason that these three were selected for this, please make this clear.

From the tittle it seems that work focuses only M. hapla, however effects on Fusarium oxysporum, F. graminearum or Rhizoctonia solani were also observed. I would recommend addressing this in the tittle of the manuscript.

How many M. incognita J2 specimen were used as inoculum (per one well) under in vitro conditions?

Was bootstrapping conducted while phylogenetic tree was constructed? Add information on that.

What is the commercial potential of your results, add information into discussion on that.

Author Response

Response to Reviewer 4

Comments and Suggestions for Authors

Fine article with logical concept and clearly formulated results, I can recommend its publication after some minor adjustment:

What was the unique scientific equipment used (lines 77 – 79), please write more about it.

We decided against using the word “unique” just in case. We believe it is state-of-the-art, new generation equipment that includes a complex of fermenters, microscopes, colony counters, freeze-drying system and other devices.

We have renewed the link to the detailed information. https://ckp-rf.ru/usu/671367/

Why were obtained genome sequences from three bacterial strains; what was the reason that these three were selected for this, please make this clear.

These three strains are the most extensively researched among those presented in Table 1.

From the tittle it seems that work focuses only M. hapla, however effects on Fusarium oxysporum, F. graminearum or Rhizoctonia solani were also observed. I would recommend addressing this in the tittle of the manuscript.

The core object of study is the influence of bacteria on RKN. We did not think it reasonable to make the title longer.

How many M. incognita J2 specimen were used as inoculum (per one well) under in vitro conditions?

100 M.incognita J2 specimen were used .

Was bootstrapping conducted while phylogenetic tree was constructed? Add information on that.

The level of support for internal branches was assessed using the Bayesian test in 88 PhyML (mentioned in lines 88-89)

What is the commercial potential of your results, add information into discussion on that.

The data obtained in this and further research will contribute to selection and comprehensive study of new biocontrol strains. They may form the basis for plant protection against plant parasitic nematodes as alternative control measure. This also implies serious commercial potential, as the demand for such formulations is growing while global supply remains insufficient.

We have included an answer to this question in Discussion (last paragraph) 

We are very grateful to the reviewer for his/her comments and proposal.

Round 2

Reviewer 3 Report

The Authors answered to my questions, but some of the answers are not followed by the addition of the information I requested. However, the other two reviewers were more positive than me, so that I think the paper can be finally accepted in the present form.